# Does Trypsin Oral Spray (Viruprotect^®^/ColdZyme^®^) Protect against COVID-19 and Common Colds or Induce Mutation? Caveats in Medical Device Regulations in the European Union

**DOI:** 10.3390/ijerph18105066

**Published:** 2021-05-11

**Authors:** Suzy Huijghebaert, Guido Vanham, Myriam Van Winckel, Karel Allegaert

**Affiliations:** 1Independent Researcher, 1310 La Hulpe, Belgium; OralMedDevs@gmail.com; 2Department of Virology, Institute of Tropical Medicine, 2000 Antwerp, Belgium; GVanham@ext.itg.be; 3Department of Paediatrics, Ghent University Hospital and Ghent University, 9000 Ghent, Belgium; Myriam.vanwinckel@uzgent.be; 4Department of Development and Regeneration, KU Leuven, 3000 Leuven, Belgium; 5Department of Pharmacy and Pharmaceutical Sciences, KU Leuven, 3000 Leuven, Belgium; 6Department of Clinical Pharmacy, Wytemaweg Hospital Pharmacy, 3075 CE Rotterdam, The Netherlands

**Keywords:** trypsin, medical device, oral spray, common cold, COVID-19, safety, mutations

## Abstract

Background: nasal or oral sprays are often marketed as medical devices (MDs) in the European Union to prevent common cold (CC), with ColdZyme^®^/Viruprotect^®^ (trypsin/glycerol) mouth spray claiming to prevent colds and the COVID-19 virus from infecting host cells and to shorten/reduce CC symptoms as an example. We analyzed the published (pre)-clinical evidence. Methods: preclinical: comparison of in vitro tests with validated host cell models to determine viral infectivity. Clinical: efficacy, proportion of users protected against virus (compared with non-users) and safety associated with trypsin/glycerol. Results: preclinical data showed that exogenous trypsin enhances SARS-CoV-2 infectivity and syncytia formation in host models, while culture passages in trypsin presence induce spike protein mutants. The manufacturer claims >98% SARS-CoV-2 deactivation, although clinically irrelevant as based on a tryptic viral digest, inserting trypsin inactivation before host cells exposure. Efficacy and safety were not adequately addressed in clinical studies or leaflets (no COVID-19 data). Protection was obtained among 9–39% of users, comparable to or lower than placebo-treated or non-users. Several potential safety risks (tissue digestion, bronchoconstriction) were identified. Conclusions: the current European MD regulations may result in insufficient exploration of (pre)clinical proof of action. Exogenous trypsin exposure even raises concerns (higher SARS-CoV-2 infectivity, mutations), whereas its clinical protective performance against respiratory viruses as published remains poor and substandard.

## 1. Introduction

During the COVID-19 pandemic, numerous oral and nasal sprays have been investigated and promoted, each claiming to protect against viral infection and COVID-19 in particular. A recent review identified 14 such sprays in development, all with ongoing studies, reflecting the appropriateness to generate data on efficacy and safety—both preclinical and clinical—before marketing such products [1]. A scoping internet search by us identified a wealth of promising treatments for the nose and throat against COVID-19, while a number of these sprays are already marketed as medical devices (MDs) in Europe. Such oral or nasal sprays may easily enter the European Economic Area (EEA) as MDs, as MDs are not conditioned the same way as medicines [2,3]. The prerequisites clearly differ from the regulatory requirements needed for registration as a medicine. Regulations for MDs are much less stringent, as there is no need to prove efficacy and safety of MDs in clinical trials prior to launch. It suffices to demonstrate that there is some clinical ‘performance’ and to generate vigilance data post-launch (Figure 1) [4,5]. There is no evaluation or approval required by the European Medicinal Agency. Only notification with a competent authority is needed in one market from the European Union, selected by the manufacturer. Notification can occur after certification of the spray by the manufacturer (package with Conformitè Europëenne—CE—marking) or by a manufacturer-selected notifying body (NB) (package with CE marking followed by the identification number of the NB). MDs are directly available over the counter (OTC) in the European Economic Area (EEA) market and promoted directly to the consumer, while claims or leaflet contents are not controlled by medicinal authorities.

We evaluated such an oral spray, containing glycerol and the serine protease trypsin, available as MD under the brand name Viruprotect^®^ or ColdZyme^®^ (further abbreviated as CZ-MD). This spray is marketed as Viruprotect^®^ by Enzymatica ltd and its licensee-holder STADA throughout most European countries [6], while as ColdZyme^®^ (CZ-MD) by Boots ltd. in the United Kingdom (UK) [7].

The spray claims to protect against respiratory viruses, and to reduce and shorten symptoms of common cold (CC). It is promoted to the family, inclusive of young children from 4 years onwards [6,7,8]. It became a leading common cold (CC) brand in the UK and Sweden (country of origin), while its sales are prohibited in Germany [9,10,11]. This is because the latter country has an additional local legislation requiring clinical proof of efficacy for MDs, which the product so far would not have delivered [11]. Since March 2020, we noted that this spray is promoted by internet sites, Facebook and direct consumer targeting—as well as in news ads—to treat or to protect against viral infections and COVID-19. For instance, the spray claims to kill or inactivate 99.9% of “the human coronavirus”, while associating its efficacy to COVID-19 by illustrations of the spiked virus and listing basic preventive measures [6,12]. Facebook promotion uses a video-clip, claiming to kill 98.3% of SARS-CoV-2, likely based on a recently published in vitro study [12,13]. It indirectly focuses also on use in infants (e.g., Bizziebaby Silver Award Winner 2019 Family Health Category), schools and athletes, while also using personal directed advertising such as on Twitter and the web (using banners/pop-ups) [8,14,15,16].

In an attempt to illustrate the relevance to fully assess the (pre)clinical safety and efficacy of such a trypsin-containing spray—similar to how medicines are assessed—before marketing, we analyzed the preclinical evidence for the antiviral claims, as well as searched for clinical trials to evaluate the efficacy of the spray to protect against CC viruses and SARS-CoV-2. We also reviewed established risks associated with trypsin use, relevant to safety. Moreover, as CZ-MD, sold by Boots in the UK [7], became a top brand marketed against common colds in the UK [9], while this country has been haunted by the surge of an easier-transmittable SARS-CoV-2 mutant [17], we searched for the effect of exogenous trypsin on viral mutant formation in vitro.

## 2. Materials and Methods

All preclinical and clinical studies identified for CZ-MD (ColdZyme^®^, Viruprotect^®^) as abstract or article, retrieved through screening on the internet (PubMed, clinical.gov database, the manufacturer’s or distributor’s site or press releases) were evaluated. For comparison of the setup and outcomes of in vitro tests of virucidal activity, all sources retrievable on the internet were used, while for independent results with exogenous trypsin in validated host cell assays, references were selected based on a PubMed search on SARS-CoV-2 combined with trypsin, resulting in 24 hits, one of which was on CZ-MD [13]. The latter hits were retained if referring to SARS-CoV-1 or other coronaviruses, while excluded if referring to absence of trypsin (*n* = 2), trypsin-like activity or other proteases (*n* = 3), review-type of publication (*n* = 3), protein analysis (*n* = 2), trypsin-1 gene variants or vaccine (*n* = 2), alpha-1-anti-trypsin (*n* = 1) or lactoferrin preparation (*n* = 1), resulting in 11 manufacturer-independent sources on trypsin. Nine more sources on the use of exogenous trypsin related to SARS-CoV(2) were identified during the literature study via citations in these articles and were tabulated as well (Appendix A) [18,19,20,21,22,23,24,25,26,27,28,29,30,31,32,33,34,35,36]. Information on mutants was generated from this set of publications. Additionally, a limited number of references were obtained from PubMed to discover the impact of exogenous trypsin in host models with other respiratory viruses, promoted by the manufacturer. For clinical evaluation: all hits on CZ-MD studies were maintained, reviewed for contents, study design, results and adverse events; wherever possible, the proportion of CZ-MD users that did not develop CC during the study period was calculated and compared with the ratio among controls (in parallel studies) or in non-users (open prospective studies), as to identify the percentage of persons protected against infection with respiratory viruses. Other clinical parameters were reviewed qualitatively with regard to their relevance and the appropriateness of the study method.

## 3. Results

### 3.1. Preclinical Pharmacology

CZ-MD claims to kill or inactivate the “human coronavirus” and SARS-CoV-2, as well as rhinovirus, adenovirus, influenza and respiratory syncytial virus (RSV), based on in vitro trypsin digests performed in laboratory tubes, and is stopped by adding a “neutralizer” (Table 1) [13,37,38,39]. The neutralizers used were the strong trypsin inhibitor Fetal Bovine Serum (FBS), Newborn Calf Serum (NBS) [13,37] or the reversible trypsin inhibitor benzamidine [38]. The neutralization step, by addition of the trypsin inhibitor, was done prior to assessing the infectivity of SARS-CoV-2 and HCoV-229E [13], five CC viruses [37,38] and rhinovirus 1A [39] in the host cells (setup illustrated in Figure 2). Such in vitro trypsin digest (or virucidal suspension test, as called by the manufacturer) may not reflect a benefit in vivo. Firstly, the trypsin digest was obtained by vortexing 1.2 mL of CZ-MD spray (4-fold the administered dose per leaflet; trypsin dose in µg/mL or units/mL not disclosed) for 20 min to accelerate the trypsinization—thus reaching a dose concentration and acceleration of trypsin activity difficult to reach in the throat. Secondly, trypsin “neutralizer” cannot be added in the mouth prior to contact of the spray with the host’s mucosa, where moreover saliva will further interfere with trypsin activity (neither assessed).

On the contrary, the 19 independent studies that we reviewed (Appendix A) consistently showed that exogenous trypsin added to host cells in presence of SARS-CoV-1 (*n* = 3), SARS-CoV-2 (*n* = 10) and other coronaviruses (*n* = 7) does not annihilate nor stop, but facilitates priming of the spike and/or cell entry, and enhances virus infectivity, cell fusion and syncytia formation in vitro [18,19,20,21,22,23,24,25,26,27,28,29,30,31,32,33,34,35,36]. The effects were obtained in the ng to µg trypsin/mL range, so likely at lower (and thus relevant) concentrations than obtained with the 1.2 mL volume used in the manufacturer’s virucidal suspension test. Appendix A summarizes detailed information on the assays and methods applied, and the key findings of the studies [18,19,20,21,22,23,24,25,26,27,28,29,30,31,32,33,34,35,36]. Virus resistance to trypsin, virus survival and enhanced or sustained cell invasion and syncytia formation with trypsin were also reported in host cell assays with rhinovirus, RSV and influenza [40,41,42,43,44,45].

### 3.2. Efficacy

There was no evidence of efficacy of trypsin and there were no planned or ongoing clinical trials identified in trials databases (ClinicalTrials.gov or EUDAMED). The % participants that were CC-free (so not developing CC and protected during the study period against respiratory virus) varied between 9–39% among Viruprotect^®^ users, lower than in non-users (30.5–63%).

The two largest studies, one comparing with placebo (protocol NCT03794804) [46], the other one comparing with a control group (NCT03831763 [47]) did not result in significant differences between treatment groups with regard to prevention of CC (information based on press-releases; actual data not released by the manufacturer) [48,49]. Only recently, a press release by Enzymatica reports that the first study assessing also the quality-of-life in 10 study centers in Germany in the spring of 2019 (701 persons enrolled) were positive [50], be it that the full report of the clinical data is not yet available. The company press release reports that with the Wisconsin Upper Respiratory Symptom Survey (WURSS-21) Quality of Life (QoL) domain and Jackson score, Enzymatica was able to document a slightly faster recovery with CZ-MD, i.e., symptoms and complaints affecting the quality of life were shortened by about half a day.

At the end of the study, a larger proportion in the CZ-MD group (70.6%) rated the effectiveness of their treatment as “very good” or “good”, as opposed to 60.1% in the placebo group (*p* < 0.05). The weblink provided with the press release on more information about the study results only leads to the protocol, without posted results [46]. The other prospective open study, including a control group, reports only data in 267 of the 400 targeted participants per protocol [47,49]. The absence of baseline information on the participants enrolled in both treatment groups (even their number per group is missing) precludes evaluation of the clinical significance of the manufacturer-claimed improvements in quality of life (expressed as area-under-the-curve (AUC), symptoms and rescue medication use during the first week [49].

In a previous randomized, double-blind placebo-controlled clinical study (referred to as ENZY-002 or COLDPREV II, protocol NCT02479750, 88 subjects, both men and women, were experimentally inoculated (infected) with the cold virus (Rhinovirus 16) [51,52]. The primary endpoint, reduction of the total viral load in the throat, and the secondary endpoint, reduction of the number of days with common cold symptoms, did not show significant differences between CZ-MD and placebo [51,52]. The full report was not released.

In a smaller open-label study in athletes, the proportions of virus-protected persons per treatment group were not released, but there was no significant difference in number of CC episodes/person between the CZ-MD and control group [53]. The study mixing (unweighted) data obtained from winter seasons over 2 years claimed a reduction of 3.5 days with CZ-MD versus the controls. However, patients in the control group felt less need for using OTC medication than in the CZ-MD group (38 versus 48%, not significant), undercutting the relevance of the findings. Moreover, the reduction in training load during CC, the return to normal training and the total number of training days were not significantly different between the groups.

Two smaller studies allowed us to calculate the % of users and non-users protected against virus [54,55]. In the small pilot study in healthy volunteers (ENZY-001 or COLDPREV, performed in 2013) using the same protocol as COLDPREV II (protocol registered in 2015 and adapted twice post-trial in 2019 and 2020 [54,56,57]), the spray use before and after a nasal challenge with Rhinovirus-16 was assessed: the protective effect of 24-h pretreatment with CZ-MD, 6 times a day, prior to the standardized viral challenge, was compared with a placebo. More CZ-MD users became qPCR positive (20/23; 83%) and symptomatic (21/23; 91.3%) compared with placebo (16/23 (69.5%) [54].

Viral load and symptoms, assessed in the subsequent days, were collected as AUCs starting only 72 h after the viral challenge (so omitting the data of the first 3 days). Hence, although the manufacturer claims shorter and less symptoms and a reduced viral load, the selective choice of data restricted over a part of the study period and moreover limited to 21 CZ-MD versus 16 placebo participants, does not reliably allow such conclusions [54]. As mentioned above, the results were not confirmed in the larger placebo-controlled study COLDPREV II [51].

The other study reports on an open survey in 113 caregivers active in elderly care: based on the analysis of the e-data responses by the participants, it could be deduced that 51/71 users (72%) contracted a cold, in comparison to 15/40 non-users, as reported in the discussion (37%) [55]. Vice versa, two more open studies in athletes revealed similarly low protection rates with Viruprotect (38–39%) [58,59]. Thus, based on these trials, the calculated protection among users varied between 9%-39% versus 31%-63% among non-users.

At last, the manufacturer reports results on his internet site, referring to a survey in 24 children over 4 years and 26 adolescents: despite the open design, only one third of the children’s parents responded they thought that CZ-MD had protected their children from catching CC, while 56% (entitled as a “majority” by the manufacturer) believed the symptoms had become milder. The outcome for the parameter “number of days with symptoms” was not released. Results were promoted by the manufacturer as ‘Shorter colds and milder symptoms among children who use ColdZyme^®^’ [60]. Table 2 lists all studies identified with CZ-MD and the calculated proportions protected during the study.

### 3.3. Safety

Isolated adverse effects reported in the studies included labial Herpes simplex infection and urticaria, not related to the MD [54] and unsettled stomach, potentially related [53]. In the prospective trial NCT03831763 reporting on 267 of the 400 targeted participants, the manufacturer’s press release discloses: “no side effects attributable to ColdZyme were reported”.

Yet, no information in clinical trial protocols appear to have addressed the potential health hazards that were identified from known effects of trypsin/glycerin: these include (1) increased infectivity of respiratory viruses on trypsin exposure (as seen with trypsin in host cells in absence of trypsin inhibition), not only for SARS-CoV-2 (Appendix A, [18,19,20,21,22,23,24,25,26,27,28,29,30,31,32,33,34,35,36]) but also for influenza and the Respiratory Syncytial Virus (RSV) [40,42,43], (2) induction of irritation and constriction of airways by inhaling glycerin/trypsin [61,62], and (3) epithelial damage and/or digestion of the protective mucin layer by trypsin possible at the level of labial, pharyngeal, oesophageal and gastric mucosa [63,64,65], (4) so trypsin/glycerin possibly contributing to hoarseness, herpes labialis eruption, gastro-esophageal reflux disease, and gastric or respiratory problems, such as occasionally reported in clinical studies or rather assessed by the manufacturer as a CC symptom [66,67,68,69,70,71,72,73,74]. Leaflets mention a caution for allergy, while it is asked not to breath during spraying, “because aspiration can cause transient-asthma-like symptoms, such as cough and hoarseness” [7]. Hoarseness was a frequent symptom reported (but assessed as CC symptoms) among 5 of 8 symptomatic athletes [59].

### 3.4. Mutant Formation

The literature analysis in Appendix A also revealed several studies indicating that serial passage of virus in cell culture in the presence of trypsin induced mutations, identified in spike (S) and other proteins of SARS-CoV-2 and other coronaviruses [28,31]. Mutants were described for their impact in presence versus absence of trypsin. Some mutants were reported to have to rely on trypsin for cell-cell fusion, or to be boosted by trypsin [27,29], while the mutants in the spikes were often mentioned to evade trypsin-dependence [20,35]. While small mutation in virus appear to be common upon passages of coronaviruses in Vero cells, the findings suggest that trypsin may facilitate their formation and lead to mutant variants with distinct clinically relevant properties, facilitating infectivity: while for instance a spike gene mutants were common in in serum-free cultures of SARS-CoV-2 in presence of trypsin, these were not detected during the propagation in TMPRSS2 expressing Vero-TMPRSS2 cells [28].

Relevant mutants of SARS-Cov2 have been described in the UK, Sweden and South Africa, countries where also a successful launch of the spray has been reported [17,75,76,77,78,79]. Sweden already identified rare mutations in the spike protein following the first wave, with 14 variant sites at ≥5% frequency in the Swedish population, while Sweden currently steps up all efforts to map mutated strains [75,76,80]. The E484K mutation is found in the UK (often referred to as the British variant [17]. While this association obviously is no a proof of causal relationship, it warrants further study.

## 4. Discussion

Analyzing the currently published (pre)-clinical evidence of a trypsin containing nasal spray to prevent CC, we came to the conclusion that exogenous trypsin exposure even raises concerns (higher SARS-CoV-2 infectivity, mutations), whereas its clinical protective performance against respiratory viruses as currently accessible in the public domain remains poor and substandard.

Trypsin is a serine protease that effectuates digestion (hydrolysis, degradation) of most proteins or peptides, be it viral, cellular, or exogenous peptides: it is commonly used to breakdown of proteins for protein analysis, to activate proenzymes of proteases, or to dissociate, harvest and resuspend cells from cell culture dishes [81].

Firstly, no independent evidence was found that trypsin inhibits in vitro replication of respiratory viruses. While the manufacturer’s studies of CZ-MD claim antiviral efficacy of trypsin against respiratory viruses, the current analysis provided evidence that exogenous trypsin rather enhances the infectivity of SARS-CoV-2 and many other respiratory viruses in host cells in vitro, thereby enhancing tissue damage or syncytia formation (Figure 1, Appendix A). Common virology laboratory setups using trypsins with respiratory viruses in presence of host cells rather show an untoward opposite effect, as recently also pointed out by others reacting to the Facebook promotion of the CZ-MD spray [82]. Apart from the overwhelming evidence from independent literature, it is noteworthy that 25 host cell lines were validated for viral growth of SARS-CoV-2 in absence and presence of trypsin: trypsin presence was not reported to reduce or induce lack of detection of infectivity [83]. We conclude that the positive results of CZ-MD on viral killing are due to the peculiar laboratory test setup, integrating a neutralizer step with trypsin inhibitor, added before exposure of the diluted reaction mixture to host cells (Table 1).

Additionally, for the other coronaviruses listed in the Appendix A, none of the independent studies showed a potential benefit of exogenous trypsin added to cell cultures for inhibiting viral invasion. Moreover, our additional searches on other respiratory viruses supported that—opposite to the desired benefit—trypsin is typically added to induce cleavage and activation of viral proteins, and/or to promote viral growth in cells, such as for RSV [40], influenza virus [42,43] and human metapneumovirus [83]. Furthermore, conformational dynamic studies of the SARS-CoV-2 spikes confirmed a role of trypsin in activating the spikes, when opening in response to human Angiotensin-converting enzyme 2 (ACE2), its impact allowing easier human ACE2-bound spike conformation [84]. Further, endogenous trypsin has been associated with pathogenicity and multi-organ failure in viral infection, such as proposed in the “influenza virus–cytokine–trypsin” cycle, involving the up-regulation of trypsin through pro-inflammatory cytokines, with the potentiation of viral multiplication in various organs [85]. How adding exogenous trypsin of a mouth spray would act differently on viral replication and tissue penetration in the oropharyngeal cavity remains elusive, further underlining the unknown risk-benefit ratio of this spray and an underexplored mechanism. Clinically, rather a role for ‘anti’-trypsin (alpha-1-anti-trypsin) has been proposed in COVID-19 [86].

Secondly, while the manufacturer claims protection with the spray against respiratory viruses, our evaluation of the numerical data (calculating the % of protected persons among users versus non-users) as reported in the clinical study reports neither revealed efficient protection of the human host against CC and COVID-19. In COVID-19, neither clinical data nor ongoing studies were identified. With regard to other clinical parameters, these were assessed in CC, such as viral load, symptoms, number of (working/exercising) days lost and quality of life: the data were generally of poor quality and not corrected for relevant confounding baseline values. Additionally, the clinical (baseline) information that is needed for adequate interpretation of the results was usually missing. To note: the rhinovirus viral load, documented in a pilot study and leading to the claim of protection, was only assessed as area-under-the-cure (AUC) over the second part of the treatment period; no individual daily viral loads were reported from the start following the standardized nasal challenge with rhinovirus [54]. By omitting the data during the first 3 days, outcomes may have unrightfully favored CZ-MD, as trypsin spray users may have recieved higher initial viral loads in the first three days, as trypsin facilitates viral cell entry in vitro according to our analysis. The fact that fewer persons on placebo developed CC after the rhinovirus challenge corroborates the findings that trypsin rather facilitates viral cell entry. The results of the second larger challenge study (per press release negative for all outcome parameters) were not published [52]. It remains to be noted that the viral load is not predictive of outcomes of CC, but rather the immune response determines the presence and evolution of symptoms [87]. Moreover, it has been shown that viral copies can be detected in nasal lavage over 15 days after Rhinovirus-16 challenge, with early, late and non-resolvers, thus invalidating the clinical usefulness of a cumulative viral load over only evaluated over an isolated short period of time after the viral challenge [88].

Overall, our evaluation also reflects the current poor quality of the clinical data of MDs, in line with former findings, as well as of the reporting in the public domain, escaping peer review [2,89]. As illustrated in Table 1 and Table 2, many (pre)clinical study data remained unpublished or were solely and selectively available on the manufacturer’s internet site. Lack of efficacy and negative outcomes were found in independent press releases, while positive excerpts were commonly found in press releases (Table 2) [11,48].

Our searches identified various potential safety hazards associated with glycerol and trypsin, such as tissue digestion, bronchoconstriction and mutant formation. Data in the public domain did not address explicitly these established health hazards. Normally, a clinical trial consent form needs to inform patients upfront of known potential health hazards of the product under investigation and optimally, and the trial needs to actively address such established safety issues [90]. If there was any mention, risks such as bronchoconstriction or hoarseness were rather reported as the consequence of incorrect use of the spray by the user in some but not all leaflets or somewhere on the product’s internet site. Hoarseness was rather approached as a clinical symptom in the clinical studies [54,59]. Strikingly, albeit not reaching statistical significance, OTC-rescue medication was felt needed in more CC episodes on CZ-MD than in the control group, debunking the spray’s benefits for treatment of CC in one study, while another open prospective study claims 23% reduced use of rescue medication with CZ-MD further fueling inconsistent findings [49,53]. No long-term studies were found addressing the safety of continuous (prophylactic) 30-day use, set as a limit in the leaflet.

Finally, our analysis raised concerns with regard to facilitation of mutations: several in vitro studies revealed that serial passage in cell cultures in the presence of trypsin is associated with mutations of the spike proteins of SARS-CoV-2 and other coronaviruses [28,31]. Although mutations are common among coronaviruses, the data suggest that coronavirus mutants may try to escape inappropriate digestive effects of trypsin on their spikes or nucleocapsides. Although speculative, the association of appearance of dominant mutants of clinical relevance in countries where the spray has been successfully launched, raises the question whether—beyond coincidence—the use of trypsins for CC may have played play a role in mutant generation of SARS-CoV-2. The spray was developed as MD by the Swedish manufacturer Enzymatica, where it is the third leading CC brand [10]. Sweden reported on the appearance of relevant mutants in the first wave, while the UK-variant also appeared in a market in whom the sales developed very well [10,76,77]. In the UK, the spray has been widely promoted to protect against viruses for cold and flu symptoms on Netdoctor (first spray recommended among the “5 best cold and flu treatment products”) [9]. It has also been launched in South Africa since spring 2019 [79]. Intriguingly, trypsin is also commonly marketed in India in various formulations as anti-inflammatory and healing agent for pain relief, sometimes in combination with non-steroidal anti-inflammatory drugs or chymotrypsin [91,92].

The spray is—in contrast—forbidden for sale in Germany as sole country in the EU. In other countries, the impact of CZ-MD is unknown, while the company uses person-targeted promotion via Facebook, popping up on the screen, as reported and experienced by some co-authors during internet search for COVID-19 [8,12,14,15,16]. Thus, although the current findings only allow hypothesis generation at this stage, it is tempting to implicate a role of trypsin spray use in the resurgence of SARS-CoV-2 mutants. Nevertheless, the literature data about the effects of trypsin on corona mutants calls for an urgent pharmacovigilance evaluation of the spray. Finally, the current case study is not a standalone effort, as McCullough recently summarized the barrier therapies supporting the biology of mucosal barrier-medical devices for common clinical mucosal disorders, either CE labelled, or licensed by the Food and Drug Administration [93]. The author hereby explicitly explored the available evidence on the physical engagement of a barrier therapy and its subsequent efficacy, and generated evidence on efficacy of some of these products (based on a standard of fitness for mucosal barrier therapies), but not for a trypsin containing formulation intended for mucosal healing [93].

## 5. Conclusions

This case study with a trypsin spray reveals the shortcomings of EU MD regulations (MDRs), as a consequence of insufficient independent assessment of substance-based MDs in the EU. We observed poor (pre)clinical viral (load) assessments allowing the manufacturer to claim protection against COVID-19 that cannot be confirmed by information retrieved in the public domain. Our study also revealed poor clinical trial design and analysis, while we could not identify active monitoring of potential safety hazards associated with trypsin. With the poor clinical data available for expert evaluation, the real life effect of this product remains hard to predict. It is currently also impossible to obtain vigilance data because the product is sold OTC and also through internet pharmacies, supermarkets, warehouses and person-targeted internet promotion. An urgent evaluation in the population is needed to exclude possible adverse effects of this trypsin spray on viral transmission, susceptibility/infectivity and mutation. This also leads back to the essence of the problem: why are medical devices intended for repeated oral or nasal administration not regulated like medicines?

## Figures and Tables

**Figure 1 ijerph-18-05066-f001:**
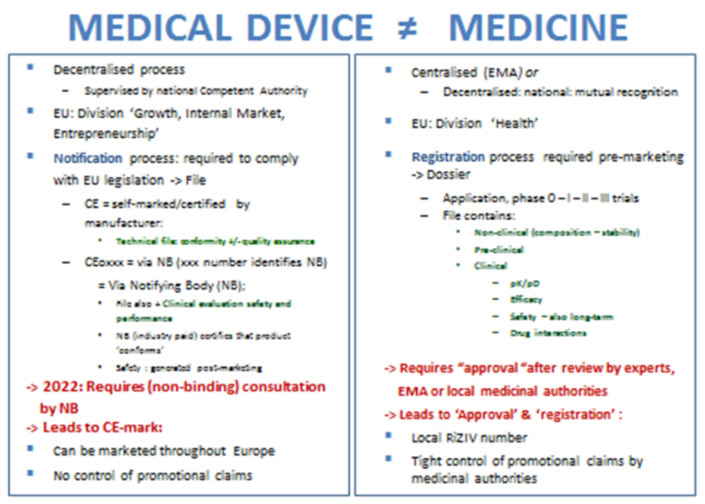
Differences in regulations between medical devices and medicines in the European Union (definition of performance: “the ability of a device to achieve its intended purpose as stated by the manufacturer”; CE = Conformitè Europëeenne; NB: Notifying body; pK/pD = pharmacokinetics/pharmacodynamics); EEA = European Economic Area. EMA = European Medicines Agency; EU = European Union [3,4,5].

**Figure 2 ijerph-18-05066-f002:**
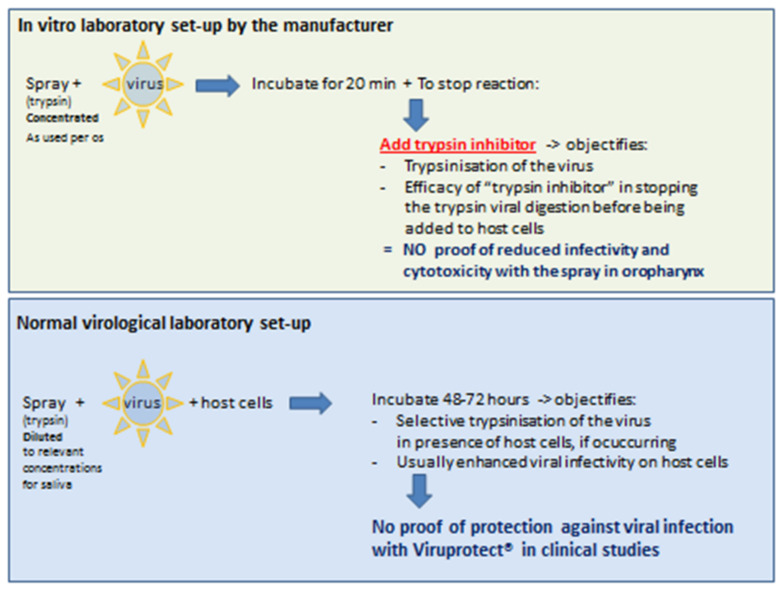
Differences in in vitro laboratory setup, as used by CZ-MD mouth spray, compared to the common approach to assess viral infectivity. (Top) Setup by the manufacturer (see Table 1 for sources [18,19,20,21,22,23,24,25,26,27,28,29,30,31,32,33,34,35,36]). (Bottom) Setup of common host cell models (see Appendix A).

**Table 1 ijerph-18-05066-t001:** Virucidal suspension tests for ColdZyme^®^ or Viruprotect^®^ (CZ-MD), nasal spray with cod trypsin.

Virus Assessed	In Vitro Virucidal Suspension Test, Results	Model Specifics + Trypsin Inhibitor (TI) Used
SARS-CoV-2 and HCoV-229E [13]	CZ-MD inactivated SARS-CoV-2 and HCoV-229E in vitro by 98.3% and 99.9%, respectively.	Direct mixing of CZ-MD formulation with virus. Reaction mixture was mixed with equal volume of “neutralizer” before dilution and culture with host cells in MEM in presence of TI: TI used: for SARS-CoV-2: 10% NCS, for HCoV-229E: 10% FBS
Common cold (CC) virus [37,38]	CZ-MD inactivated rhinovirus 1A: 91.7% (1.08 log10); rhinovirus 42: 92.8% (1.14 log10); hIV virus H3N2: 96.9% (1.51 log10); RSV: 99.9% (2.94 log10); adenovirus T2: 64.5% (0.45 log10); human coronavirus [38].	Direct mixing of CZ-MD formulation with virus. Trypsin was neutralized prior to dilution and exposure to host cells. TI used: FBS
Rhinovirus 1A [39]	Trypsin delayed Rhinovirus-1A infection of RD cells in vitro by 2 days.	Direct mixing of cod trypsin with virus. Trypsin was neutralized prior to dilution and exposure to host cells TI used: benzamidine

CZ-MD = Coldzyme medical device; Foetal Bovine Serum = FBS; Newborn Calf Serum = NCS; human Influenza A = hIV; Respiratory Syncytial virus (RSV); RD-cells: presumably a human rabdosarcoma cell line (abbreviation not specified in abstract; MEM = minimal essential media; TI = trypsin inhibitor [39]).

**Table 2 ijerph-18-05066-t002:** *Does Viruprotect/ColdZyme (CZ-MD) protect ‘against’ viruses?* Clinical outcome as derived from studies assessing CZ-MD and if data available, compared with placebo, controls or non-use.

Clinical Study	% Users Calculated or Rated as Protected *	Source Document
**A. Viruprotect^®^/ColdZyme^®^ versus placebo-users, controls or non-users**		
**ENZY-001** = **COLDPREV** Double-blind study in healthy subjects, following a standardized nasal rhinovirus challenge: 1 dose (=2 puffs) 6× daily, 24 h use prior to viral challenge (Day-1) + 11 days after challenge (Day 0–10) *n* = 23 CZ-MD *n* = 23 Placebo	**Protective efficacy of 24 h prophylaxis:** - **CZ-MD:** 9% (19/23 become qPRC(+) and 21/23 become symptomatic) - **Placebo:** 30.5% (16/23 become qPRC(+) + symptomatic) Manufacturer claims 3.5 days less symptoms and a lower total viral load. Yet, assessment as AUC_Day3–10_ removes all participants not developing symptoms and data during the first 3 days of use after viral challenge (Day 0–1–2); no individual data plotted over time.	Protocol: ClinicalTrials.gov NCT02522949, performed in 2013, registered post-trial in 2015, and changed in 2019 and 2020 [56,57] Publication: Clarsund et al. 2017 [54]
**ENZY-002 = COLDPREV II** Double-blind study with the same design as **COLDPREV I** but with more test persons (*n* = 88)	**CZ-MD = Placebo** —Data unavailable in public domain— Press release mentions: ‘The primary endpoint, reduction of the total viral load in the throat, and the secondary endpoint, reduction of the number of days with common cold symptoms, did not show significant difference between ColdZyme and placebo.’ [52]	Protocol: ClinicalTrials.gov NCT02479750 [51] Study results unpublished Enzymatica press release [52]
**NCT03794804** Double-blind randomised placebo-controlled study (*n* = 701)	**CZ-MD = Placebo** —Data unavailable in public domain— Consequence: sales in Germany forbidden [11] New press release by Enzymatica claims shortening of CC episode by 0.5 days [50].	Protocol: ClinicalTrials.gov NCT03794804 [46] Study results unpublished Independent press release [48] Company press release [50]
**NCT03831763** [47] (Prospective) single (investigator)-blinded, randomized, parallel study: CZ-MD versus no intervention: (*n* = 400 targeted, 267 analyzed) —Start study 2018–Status: completed 2018—	**Undisclosed**:—Data unavailable in public domain—; Current abstract in press release by manufacturer: reveals cumulative 7-day AUC’s for total Jackson score and quality of life in 267 of 400 targeted participants (however, no numbers, no baseline data, nor reasons for exclusion). Manufacturer claims 7-day AUC to be more improved with CZ-MD, as well as significant shortening of disease duration and 23% less subjects using symptom-relieving medication throughout first week [49]	Enzymatica press release [49]
**Observational study**, 6 months prophylaxis, prophylaxis, health care staff: comparison of sick leave with the previous year (*n* = 111): *n* = 71 users; *n*= 40 non-users	**Data derived from discussion section of the paper and e-data** - **CZ-MD:** 28% (16/72) - **Non-users:** 69.5% (25/40)	Clarsund et al. 2017 [55]
**Observational study**, 2 x 3-month periods, prophylaxis in endurance athletes: Dec 2017-Feb 2018 + Dec 2018-April 2019 (compliance enhanced to 6×/day in 2018–2019) *n* = 62 CZ-MD *n* = 61 controls —4–10 h of training/week—	**CZ-MD = Control group** for number of episodes/person (No data on the proportions of persons per treatment group). Other parameters (e.g., claimed reduction of the CC episode by 3.5 days with CZ-MD versus controls) are not corroborated by the fact that less subjects in the control group felt need for rescue medication than on CZ-MD (38% vs 48%). No significant effects on training-related outcomes. Data on CC episodes were not or not consistently corrected for compliance versus baseline variables and different treatment periods.	Davison et al. 2019 [53]
**B. Data restricted to users of Viruprotect^®^/ColdZyme^®^**		
**Observational study** in athletes *n* = 11 biathlon, 3 months *n* = 29 ice hockey, 12 months *n*= 20 hand ball, 6 months	**CZ-MD:** total: 13/33 = 39% → 1 op 11 (=9%) → No data → 12 in 20 (=60%)	Clarsund et al. 2017 [58]
**Observational study** in 13 athletes, 3 months	**CZ-MD:** 5 in 13 (=38%)	Blom et al. 2018 [59]
**Observational study** 24 children 4–12 years 26 adolescents 12–19 years	**CZ-MD:** ~1/3 (=33%) of the parents ‘*think* it could have a protective effect’	Enzymatica, press release [60]

* Calculation of protection = number of persons not developing Rhinovirus infection or CC symptoms, derived from the total number of subjects treated minus those developing symptoms. AUC = Area-Under-the-Curve; CC = common cold.

## Data Availability

The data presented in this study are available based on a reasonable request form the corresponding or the first author.

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
