# Peer review of "Does Trypsin Oral Spray (Viruprotect®/ColdZyme®) Protect against COVID-19 and Common Colds or Induce Mutation? Caveats in Medical Device Regulations in the European Union"

_ijerph, 2021, doi:10.3390/ijerph18105066_

Round 1
Reviewer 1 Report
In the present manuscript, the authors suggest implementing spray or aerosol with the possible infection potential with viruses. However, the information is no clear, and the evidence experimental to mention is insufficient to suggest the effect of trypsin, such as catalyzed the event of entry of viruses. My recommendation to authors is to write in detail the experimental evidence or virological trial that supports this situation; the tables' information does not specify the test or methods used to support the findings.
Author Response
We thank the reviewer for the valuable suggestions.
Perhaps the reviewer has somewhat misread our ‘clinical message’, as we intended to highlight the relevance to fully access the (pre)clinical safety and efficacy data, similar to how a medicine should be assessed, so shifting away from the medical device regulations currently ‘standing’ in the EU. We have further highlighted this in the revised version of the paper (cfr track changes version).
Along the same line, we have truly considered the recommendation to add more details on the methodology described. To do this, without obstructing the current text flow, we have added the detailed information on the revised version of the supplemental table S1 (the additional information is highlighted in red to facilitate revision, with subsections on assay/methods and results).
Reviewer 2 Report
I strongly support that the manuscript (ID: ijerph-1172001) written by Suzy Huijghebaert, Guido Vanham, Myriam Van Winckel and Karel Allegaert entitled “Does Trypsin Oral Spray (Viruprotect®/ ColdZyme®) Protect Against COVID-19 and Common Colds or Induce Mutation? Caveats in Medical Device Regulations in the European Union” will be published in the scientific journal “International Journal of Environmental Research and Public Health“.
The reason is as follows: It is extremely important to publish such serious scientific data as soon as possible in order to prevent that too many non-scientific publications about such topic are dominating at the moment. The only point that the authors should consider. This is an extremely dynamic field and before the final draft is published it has to be carefully checked whether additional actual references have to be added.
Author Response
We agree that the COVID prevention strategies based on spray or similar approach is a very dynamic field. However, we had the intention to focus on the presence of trypsin in these sprays.
As requested, we have reconducted a search on PubMed with ‘trypsin spray’ and could not find new data compared to the paper as submitted on 21.03.2021. Along the same way, we also screened for potential preprints (bioRxiv) that resulted in one hit, be it off target to our research question (hormon induced male infertility). Finally, we have search Google Scholar with the same search term, and this resulted in one additional, relevant hit (Mc Cullough RW. Barrier therapies supporting the biology of the mucosal barrier-medical devices for common clinical mucosal disorders (Transl Gastroenterol hepatol 2021).
We have added this reference to the paper in the discussion section of the revised version.
Round 2
Reviewer 1 Report
The authors improved a new version of the manuscript and followed the recommendations. For which I consider that the article is accepted in this version.